# Virtual Extensive Read-Across: A New Open-Access Software for Chemical Read-Across and Its Application to the Carcinogenicity Assessment of Botanicals

**DOI:** 10.3390/molecules27196605

**Published:** 2022-10-05

**Authors:** Edoardo Luca Viganò, Erika Colombo, Giuseppa Raitano, Alberto Manganaro, Alessio Sommovigo, Jean Lou CM Dorne, Emilio Benfenati

**Affiliations:** 1Department of Environmental Health Sciences, Istituto di Ricerche Farmacologiche Mario Negri IRCCS, 20156 Milano, Italy; 2KODE Srl, 56122 Pisa, Italy; 3Methodological and Scientific Support Unit, European Food Safety Authority, 43126 Parma, Italy

**Keywords:** read-across, carcinogenicity, botanicals, software

## Abstract

Read-across applies the principle of similarity to identify the most similar substances to represent a given target substance in data-poor situations. However, differences between the target and the source substances exist. The present study aims to screen and assess the effect of the key components in a molecule which may escape the evaluation for read-across based only on the most similar substance(s) using a new open-access software: Virtual Extensive Read-Across (VERA). VERA provides a means to assess similarity between chemicals using structural alerts specific to the property, pre-defined molecular groups and structural similarity. The software finds the most similar compounds with a certain feature, e.g., structural alerts and molecular groups, and provides clusters of similar substances while comparing these similar substances within different clusters. Carcinogenicity is a complex endpoint with several mechanisms, requiring resource intensive experimental bioassays and a large number of animals; as such, the use of read-across as part of new approach methodologies would support carcinogenicity assessment. To test the VERA software, carcinogenicity was selected as the endpoint of interest for a range of botanicals. VERA correctly labelled 70% of the botanicals, indicating the most similar substances and the main features associated with carcinogenicity.

## 1. Introduction

Historically, read-across has been routinely applied to fill data gaps and has also been used by industry to provide data within the legislation Registration, Evaluation, Authorisation and Restriction of Chemicals (REACH) in more than 25% of the registrations at a similar extent to animal experimental tests (27.1% of registrations) and at much larger extent than quantitative structure–activity relationship (QSAR) models [1]. The scientific basis of read-across lies in the idea that the property values of two similar substances are also similar. The same concept underlies the application of QSAR models, although with two major differences: QSAR models are based on a much larger population of substances, as well as numerous descriptors to characterise the chemicals. In contrast, read-across applications can involve even only one source substance to fill the data gap, which is a much easier approach to assess similarity. In addition, read-across represents an opportunistic approach, since it requires similar substances with experimental data.

Various available approaches for read-across have been considered, applying different metrics to assess similarity, namely structural similarity, physicochemical properties, toxicological and/or toxicokinetic properties, in vitro or OMICs data, and their combinations [2]. The structural similarity is the most frequent approach and has been applied historically, in a manual fashion by experts, although today, read-across can be performed in an automated manner using specific open access or commercial software. Overall, structural similarity is measured using tools of different complexity, such as structural keys, fingerprints, molecular descriptors and quantum similarity [3,4,5,6].

Most often, physicochemical properties are calculated using molecular descriptors or other means to represent chemical structures. The advantage of physicochemical properties over chemical descriptors is that they are intuitive and easy to understand, and are commonly used to characterise certain properties such as bioconcentration, which in many cases is closely related to logKow, the logarithm of the partitioning between octanol and water. Profilers for toxicity are often used to cluster substances with some structural features that can be identified as responsible for the observed adverse effects [7]. In this case, specific open-access software packages, such as the OECD QSAR Toolbox [8] and COSMOS [9], are available for the purpose of toxicity profiling.

Substances can also be characterised based on the presence of common metabolites or metabolic pathways as part of the toxicokinetic profile [10]. This evaluation of the similarity goes beyond the individual substance and anticipates the transformation of the substance into a second one, which may be responsible for an observed effect. In addition, assessing similarity in other toxicokinetic properties such as absorption, distribution and excretion can also be useful to compare substances [2]. 

Experimental values using alternative experiments, such as those generated using new approach methodologies (NAMs), are being increasingly used [10,11]. Such NAM data may come from OMICs experiments or in vitro tests. This type of information is experimental, independent of the chemical structure, thus very different from the previous one. However, experiments must be run for the substances of interest, while the similarity methods based on the structure do not require experiments since the chemical structure and historical experimental results are sufficient.

One can also combine several metrics to evaluate similarity. For instance, the ToxRead and RAXpy software, both available from VEGA—www.vegahub.eu (accessed on 21 August 2022) [12,13]—employ multiple approaches for read-across. In addition to the structural similarity, ToxRead uses information characterising the toxicological profile, through structural alerts (SAs), physicochemical data and a number of molecular descriptors associated with the effect. All these properties are calculated, although experimental values may be used when available. RAXpy uses a more heterogeneous set of parameters, as input, including in vitro data as well as metabolism information [10]. 

Regardless of the approach, the key concept is to identify the most similar substances to represent the target substance. This strategy, though, may be the original sin of the strategy: the aim being to find the substance(s) representing the best target chemical; however, differences between the target and the source substances always exist. Indeed, the very word “similarity” implies that the source substances are not identical and that there are differences. The main weakness of the read-across is in fact that there may often be the suspicion that the difference between the two substances plays an important role in the effect of the target substance, and thus this aspect needs to be assessed thoroughly.

Over the last five years, the ToxDelta open-access tool (also available from VEGA) has been developed to identify and characterise such differences. The software identifies the maximum common sub-structure, and then the different components of the molecules are investigated based on the collection of fragments with known property values [14].

The present study aims to screen and assess the effect of the key components in a molecule which may escape the evaluation for read-across based only on the most similar substance(s) using a new open-access software: Virtual Extensive Read-Across (VERA), available here: https://github.com/EdoardoVigano/Virtual-Extensive-Read-Across-VERA- (accessed on 21 August 2022). VERA aims to process many possible features present in target substances, and uses similar substances with these features to assess whether any of them may be responsible for a given adverse effect, even though this feature may not be present in the most similar substances. In other words, the purpose is not to identify the most similar substances, as the key “oracles”, but to go beyond the principal read-across process.

This concept is very relevant to a range of endpoints, particularly for carcinogenicity, which is driven by complex biological processes including multiple mechanisms of carcinogenesis. In addition, investigating in vivo carcinogenicity requires several experimental bioassays which are very resource intensive including many animals, and are lengthy (two-year) and expensive studies. Hence, the use of read-across approaches as NAMs provides a potential useful tool to support carcinogenicity assessment for data poor compounds. In addition, in food and feed safety, plant ingredients include thousands of compounds, and most of them have not been tested experimentally for their carcinogenic properties. For this purpose, read-across approaches have been applied here using the VERA software for the carcinogenicity assessment of botanical compounds. 

## 2. Results and Discussion

### 2.1. Structural Alerts and Molecular Groups in VERA

The VERA tool has been developed for read-across to screen similar substances, covering different potential clusters of similar compounds. We identified a set of molecular groups (MGs), which may represent important features present in the target substance. Such MGs were used as a filter to sort out similar substances, while the VERA software searches for similar compounds containing these MGs. Then, this workflow enables the exploration of a rationale for the biological/toxicological activity related to their presence. Overall, this process allows the screening of multiple factors that may affect a biological/toxicological property, as opposed to simply identifying the most similar substances. In this context, screening such multiple factors allows read-across practice to include important missing structural or biological factors that would be otherwise ignored, assuming structural similarity as the driving process of hazardous property. Again, the key focus of the approach was to avoid wrong assignments for read-across using assessment of multiple factors, whereas simply assessing structural similarity may filter out important similar substances in the process. Since VERA relies on grouping, it is of course a valuable tool for grouping too.

Needless to say, the property itself is an important component in this evaluation. Thus, another set of molecular features is also used here to particularly search for SAs associated with the toxic effect. In other words, the overall system is composed of a set of molecular features specific to the endpoint (SAs) under consideration, and another set, which is unrelated to the endpoint: the MGs, defined only from a chemical point of view. Consequently, the general strategy is applicable to a whole range of endpoints; while the MGs remain the same, the molecular moieties associated with the adverse effect, such as the SAs, may vary. 

### 2.2. Similarity Based on the MGs

MGs are used for three key purposes: (1) clustering substances sharing the same MG and applied to SAs; (2) characterising the prevalence of active substances with a certain MG and SA; (3) measuring similarity between different substances. This is a unique procedure and provides new metrics for similarity based on the MGs.

From a theoretical point of view, more than 40 different indices can be calculated considering the presence of a molecular feature (such as a MG) in the target and the source substance. Indeed, four scenarios can be considered since the molecular features can be (a) present in both substances, (b) absent in both, (c) present in the target but not in the source; and (d) present in the source but not in the target [15]. In the approach outlined here, features absent in both substances were not considered. Furthermore, a higher weight was assigned to the MGs in common, while a much lower weight was attributed to MGs present only in the source compound (see Methods), to fulfil the aims to screen the substances indicating whether the presence of a certain MG is important for the toxicological effect under evaluation. This allowed the extraction of substances containing a certain MG providing evidence for its association with the toxicological effect. This approach also uses the open-access in silico platform VEGA, addressing global structural similarity and the associated nearest K-neighbour (K-NN) models [5], as described in Figure 1.

### 2.3. The Conceptual Workflow

The overall strategy developed here from the conceptual workflow initiates with filtering molecules according to MG_1_ (see Method), and then identifies the key factors contributing to the toxicological effect, as represented in Figure 1. In the case of carcinogenicity, this is represented by the Benigni/Bossa SAs, as implemented in Toxtree [16]. Then, the software searches for similar compounds with the same SA in the target substance. If the ten most similar substances obtained with this strategy have the same experimental value, the software classifies the target substance with a higher reliability according to the label. Otherwise, if the target does not even have one SA, the software initially focuses on the ten most similar compounds so that:

If at least seven out of ten are active, the target is classified as active, and the reliability depends on the number of substances. 

- If the software does not reach a conclusion in this preliminary phase, it assesses the hypothesis that there is a certain theoretical reason for this adverse effect—the SA or MG with the highest toxicity prevalence—a reason supported by similar substances which are carcinogenic, and whether this is true for the target substance (Figure 2). This, for instance, may be the case for a certain factor modulating or decreasing the toxicological effect, acting as an exception rule. 

- If there are known exception rules, the software also uses them, so that in practice the target substance will not be labelled as active if it contains a known exception rule. Thus, in this case, unknown molecular features are explored, acting as exception rules, and these may be applicable to the target substance. This is described in Figure 2. 

### 2.4. Solving Conflicts between Clusters

If all the clusters are demonstrated to have the same outcome, active or inactive, there is no ambiguity in the overall evidence.

When there are conflicting outcomes in two or more MGs, VERA merges all the “Active” clusters in the ACTIVE cluster and all the “Inactive” ones in the INACTIVE cluster. These two clusters are analysed by the VERA algorithm to assess the activity of the target molecule (see Methods). The VERA algorithm searches substances present in both clusters, since each cluster may represent a different process related to the property of interest, namely a process pointing towards activity, and another one pointing towards inactivity. Looking at the substances containing both the features related to activity and inactivity, one can assess which one would prevail. For such an assessment, VERA searches substances containing both factors providing a basis to see whether the substances with the different structural components (either SA or MG) are active or inactive. These substances with both structural components define the overall evidence in the case of conflicting results. The number of these substances and the consistency in their label are used to define the level of uncertainty in the outcome. 

Finally, in some cases, the target compound does not contain a SA. In this case, VERA searches for the MG with the highest prevalence of active substances, and thus, a substance may be labelled as toxic even without a known SA (Figure 3).

### 2.5. Representation of the Prevalence

VERA shows the prevalence of active substances for the SAs and MGs for the endpoint of interest and uses an internal database with experimental values. This information is useful to characterise the importance of a certain SA or MG. In some cases, the prevalence of active substances with a certain MG was shown to be higher than that from active substances in the SA. This demonstrates that MGs are useful components in the process, enabling the exploration of factors related to toxicity which may escape the common use of SAs. Appendix A lists the SAs and MGs used for carcinogenicity, and the prevalence of active substances.

### 2.6. Carcinogenicity Assessment of Botanicals

VERA was applied for the carcinogenicity assessment of botanical substances, and experimental values were found for 63 substances within the VEGA database. Then, the assessment generated by VERA for these substances was examined, and Table 1 shows the confusion matrix and the associated statistics. The accuracy, sensitivity and specificity are all close to 70%, indicating that, for this exercise, VERA provides relatively good and balanced prediction results, with the exception of 10 compounds which were not assigned since the tool recognised them as equivocal. These statistical values are good for the specific endpoint, with results similar or better compared to those reported for large collections of data [17]. We also used publicly available QSAR models (VEGA [13] and Toxtree [16], since these can be used in batch mode) for the same botanicals with experimental values, as used in the present study. Figure 4 presents the results. VERA provided the best results in sensitivity, and good results for accuracy and specificity. We notice that VERA provided values for 53 substances, and not for 63, as we commented. 

VERA has not been implemented in the first place as a predictive tool since its purpose is to perform automatised read-across, indicating possible rationales linking molecular features and toxicity (as SAs and MGs) in the target compound, and identifying the most important similar compounds associated with each of these molecular features. Thus, the user can explore different molecular features and substances representing these molecular features.

However, since VERA is an automatic tool, it can be used to generate predictions too. In this context, the authors would recommend using several predictive tools, including QSAR models predicting carcinogenicity such as those available within VEGA. Overall, VERA provides a further independent way to assess the effect, based on the experimental values of similar substances.

### 2.7. Comparison between Available Read-Across Tools and VERA

The traditional strategy for read-across is to find the substance(s) that is the closest to the target substance. These similar substances indicate values that would be the correct value for the target substance. Two main weaknesses can be pointed out in this approach: (1) To identify the most similar substance(s), one has to filter them, pruning the initial set of candidates. Thus, only the key substances are used in the final process, and the information from the other substances is not used, disregarded and considered irrelevant. However, there may be substances with lower overall similarity containing feature(s) in common with the target that can suggest a toxicological effect. (2) Classically, this selection uses similarity indices, which are often continuous, so there is an arbitrary default threshold to define the criterion for inclusion of a substance, and the results may vary depending on the value of such a threshold. 

In contrast, VERA has two main advantages compared to the traditional read-across approaches, based on finding the most similar compounds: (1) The evaluation of multiple factors impacting on the toxicological effect and characterised by molecular moieties provides a more systematic approach and allows those factors contributing to the effect to be taken into account, even if apparently of little importance. For this purpose, all similar substances containing a certain MG are assessed to check whether the exclusion of a given MG implies the loss of evidence indicating activity. Hence, the algorithm for similarity based on the MG assigns greater weight to the presence of common MGs. (2) The identification of a cluster is another advantage. Substances containing a certain MG are classified and a robust criterion for membership to the cluster is set without any arbitrary threshold for inclusion. A specific molecular feature (MG or SA) is used one by one, and this procedure has the advantage of clearly identifying the possible molecular moiety involved in the toxic property. This MG is used as a singular component, and each MG is scrutinised separately. This helps with the reasoning while increasing the confidence in the assessment, since the process is transparent, clearly traceable and testable. Conversely, the classical use of aggregated factors concurring to the definition of similarity is less transparent, and the identification of the borders for similarity is not as robust. As a comparison, the tool for similarity used within VEGA applies several indices and algorithms to capture a broad series of possibilities to evaluate structural similarity [5]. For VEGA, a quantitative value is obtained, optimised using data for millions of substances. In general, a substance with a similarity below 0.75 within VEGA is probably not relevant, while a substance with a similarity above 0.9 is highly relevant to the target compound. However, this comparison may vary in different circumstances: (a) if the molecule is small, the index declines more rapidly; (b) if there are molecular features relevant to a certain endpoint, this would not be considered. Thus, there is no absolute value for similarity, and in most cases, the similarity has fuzzy borders. Furthermore, these borders define, in principle, only a single family of substances, the similar ones. Thus, the classical strategy is appropriate to identify very similar substances, but is less robust to define the distance and the possible presence of multiple clusters of similarity. Overall, VERA copes with these above-mentioned issues and provides a univocal definition for cluster membership.

## 3. Methods

### 3.1. Molecular Groups (MGs)

A list of MGs has been identified to characterise the main molecular features present in the substance. These MGs have been extracted from the RDKit molecular groups list (https://www.rdkit.org/docs/GettingStartedInPython.html, accessed on 21 August 2022, and IstChemFeat developed by KODE chemoinformatics (https://www.kode-solutions.net/en/, accessed on 21 August 2022) [18]. An initial subset was filtered out to reduce redundancy and simplify the algorithm. Furthermore, some MGs extracted manually were identified and added to the previous list as part of the evaluation process of preferable MGs. Appendix A provides the full list of MGs.

The prevalence of MGs in the training dataset was calculated to define a hierarchy of MGs. Such prevalence calculation was a pre-requisite for cases under which no SA could be identified in the target compound. In this latter case, the MG with the highest prevalence of toxic substances was used.

### 3.2. Algorithm for Similarity

The VERA tool initiates the calculation using the VEGA software testing similarity between the target substance and those in the training set based on IstSimilarity developed by KODE chemo-informatics testing (https://www.kode-solutions.net/en/, accessed on 21 August 2022). For each target substance, the tool selects all substances in the dataset with a VEGA similarity index higher than 0.65, which is quite a low value, as we discussed above, in order to keep most substances. All charged substances and salts with a similarity index greater than 0.99 compared to another similar neutral substance are omitted to avoid redundancy in the toxicological information. Consequently, the algorithm generates a subset that contains a list of similar substances which is then used for read-across.

In addition, when at least one SA is found for the target, similar substances are selected only if they have the same SA. The tool also highlights the most likely reason for toxicity of the target according to the MG hierarchy, generated in the first step, and as for the SA, similar substances in each subset are filtered with the MG with the highest prevalence of toxic substances.

Once the main criteria for toxicity (SA or MG) have been defined, the grouping similarity index (GSI) is calculated for each target and all similar substances in related subsets as follows:(1)$$GSI=\frac{\frac{1+nMGs\in common}{ntargetMGs}}{2}-\frac{\frac{nMGsNOT\in common}{nsimilarMGs}}{8}$$

Here, “nMGs∈common” indicates the common number of MGs in the target and the similar substances; n target MGs and n similar MGs refer to the number of MGs in the target and the similar substances, respectively. Finally, n MGs NOT in common refers to the number of MGs which are not in common in the target and the similar substances. Similar substances for each target are listed in descending order of the GSI. In the denominator, we chose to disadvantage the n MGs NOT in common in the calculation of the grouping similarity index. Thus, these groups contribute 1/4 over common MGs to the grouping similarity index. For this reason, the denominator is 8 in the case of the MG NOT in common, while the value in the case of MG in common is 2.

### 3.3. Implementation of VERA

VERA establishes the possible classes of the target substance with different degrees of reliability using grouping according to MGs and SAs. The read-across prediction considers the similar substances processed for each target, as described in the scenarios below (1 to 6). If the first ten similar substances have the same experimental value, the software classifies the target substance according to the label with higher reliability. On the other hand, if the target does not even have an SA, the software focuses on the ten most similar compounds to make a prediction according to GSI, as follows:(1)If n of compounds with experimental values “Active” is greater than 8, the class of the target compound is Active***, where *** indicates a prediction with “high reliability” (thus, if n “Active” > 8: Active***);(2)If n of compounds with experimental values “NON Active” is greater than 8, the class of the target compound is NON Active**, where ** indicates a prediction with “moderate reliability” (thus, if n “NON Active” > 8: NON Active**);(3)If n of compounds with experimental values “Active” is between 6 and 8, the class of the target compound is Active** (thus, if 8 ≥ n “Active” > 6: Active**);(4)If n of compounds with experimental values “NON Active” is between 6 and 8, the class of the target compound is NON Active*, where * indicates a prediction with “low reliability” (thus, if 8 ≥ n “NON Active” > 6: NON Active*);(5)If n of compounds with experimental values “Active” is 6, the class of the target compound is Active* (thus, if n “Active” = 6: Active*);(6)In the other cases, the class of the target is “Equivocal”.

If there are not 10 similar substances or the target was defined as Equivocal, the algorithm proceeds through the process of reasoning, described below, to define the class.

The VERA tool does the grouping and read-across in three ways depending on three underlying conditions: (1) at least one SA and more MGs are present; (2) one SA and one MG are present; (3) no SA is present. 

#### 3.3.1. VERA Algorithm When at Least One SA and More MGs Are Present

Depending on the number of MGs present in the target molecule, VERA generates n clusters grouping up to six molecules, each containing the SA and one MG of the target substance. Thus, for n MGs, VERA generates n clusters if they contain at least two similar compounds. Substances in the cluster are listed according to the VEGA similarity index. Each cluster is classified as “Active” or “Inactive” depending on the number of Active or NON-Active substances in each group. Consequently, if the cluster contains at least 50% of the molecules with “Active” experimental values, the cluster is labelled ACTIVE; whereas, if the cluster contains at least 65% of the molecules with “NON Active” experimental values, the cluster is labelled INACTIVE. In the other cases, the cluster is not considered, and is labelled as “No data”. All these clusters are shown in the reasoning section and the output, and form the first grouping for the read-across. The VERA algorithm is conservative, so it is more likely to provide an outcome as “Active”. 

This method can highlight possible exception rules such as the specific MG that reduces toxic potency of the SA in the target molecule. Consequently, if one of the clusters has six similar substances all of which are non-toxic, the MG that characterises the cluster is considered a possible exception rule. If no exception rule is found, the algorithm follows the reasoning, assessing whether the toxic features prevail over those indicating no toxicity. Such toxic features involve the clusters defined by one or more MGs with a prevalence of toxic substances. Thus, all active clusters are merged into one ACTIVE CLUSTER and all duplicate substances are deleted. In the same way, all inactive clusters are combined into one INACTIVE CLUSTER and duplicate substances are removed. These clusters can contain substances with experimental data which differ from the label of the cluster.

At this point, the VERA algorithm compares the number of substances in both these two final clusters. Equations (2) and (3) give an initial prediction:

If
(2)$$\frac{nsubstances\in ACTIVECLUSTER}{nsubstances\in ACTIVECLUSTER+nsubstances\in INACTIVECLUSTER}\geq 0.75$$
or
(3)$$\frac{nsubstances\in ACTIVECLUSTER}{nsubstances\in ACTIVECLUSTER+nsubstances\in INACTIVECLUSTER}\leq 0.25$$

If Equation (2) holds true, the target substance is labelled “Active **”, where ** represents the reliability, and in this case “moderate reliability”. If Equation (3) applies, the target substance is labelled “NON Active *”, where * corresponds to “low reliability”.

In the other cases, the algorithm proceeds iteratively considering the co-presence of substances in both final clusters, depending on the percentages of Active and Not Active experimental values in each cluster. Appendix A indicates percentages and associated levels of reliability. According to this procedure, predictions can be of high, moderate or low reliability. The prediction is Equivocal when the algorithm is not able to establish the class of the target substance with adequate accuracy.

The algorithm is conservative, since different thresholds are applied for the classification of Active substances.

#### 3.3.2. VERA Algorithm When One SA and One MG Are Present

In the case of the presence of one SA and a single MG, only one cluster results. In compliance with Section 3.3.1, the cluster is labelled Active or Inactive, and the first ten similar substances described are considered for read-across predictions. In this case, if the cluster is active AND the number of Active similar substances is greater than the NON Active similar substances, the target substance is labelled as Active**. If the cluster is inactive AND the number of NON Active similar substance is greater than the Active similar substances, the target substance is labelled as NON Active*. In the other cases, the target substance is labelled as “Equivocal”.

#### 3.3.3. VERA Algorithm When No SA Is Present

The first ten similar substances are considered for the read-across prediction. In this case, the reasoning is made using, as SA, the MG with the maximum prevalence in active compounds from the training set, and the software proceeds as highlighted, as described in Section 3.3.1 in the presence of SA.

### 3.4. Data Sources

As described above, the VERA tool has been tested using carcinogenesis as the endpoint of choice, and all training datasets were retrieved from the VEGA Carcinogenesis models (CAESAR, ANTARES, ISS, ISSCAN) [19,20,21]. All data were assessed for the presence of duplicates, resulting in a final dataset of 1770 substances. The approach was then assessed focussing on a subset of botanical substances and available toxic potency values, as described above. For this purpose, botanicals were identified using the list of substances from Raitano et al. [22].

### 3.5. Model Availability

VERA is downloadable from the following link: https://github.com/EdoardoVigano/Virtual-Extensive-Read-Across-VERA-, accessed on 21 August 2022, (beta version). In the near future, it will be available also at the Zenodo site of EFSA (https://zenodo.org/deposit/7106070, accessed on 21 August 2022), and it will be implemented within VEGAHUB [13].

## 4. Conclusions

Here, a new read-across and grouping approach, together with the VERA open-access software, has been described, and the results highlight that these provide means to assess substances in a batch model through an automatic reproducible process for read-across predictions of chemical toxicity properties. It is downloadable currently as a beta version, and an updated version will be implemented in VEGA soon, offering further features. Furthermore, the VERA software has been tested for the assessment of carcinogenicity associated botanical substances and their potential carcinogenic potential, providing reliable and accurate predictions with several advantages that are indicated below.

This approach is innovative compared to the traditional one which is focused on the identification of the most similar compounds according to a single metric. VERA explores numerous clusters of similar substances, and compares the clusters to provide an overall assignment, according to a predefined workflow.

VERA can be used to explore multiple similar substances which can be assessed by expert judgement, while the toxicity effect of these substances is predicted by the software. VERA provides some useful features in that it allows the following: (1) The identification similar substances. (2) The processing of multiple features underlying adverse effects. This supports expert judgement through providing a range of features to scrutinise the hazardous properties of chemicals. (3) It provides a classification of MGs as elements of reasoning associated with toxic properties, as well as further views beyond the single key SA. Traditionally, the SA is simplistically applied to an individual toxic property and additional components are not considered. (4) It provides predictions with the associated level of reliability. 

It is foreseen that further evaluation of the VERA tool for read-across predictions of a range of endpoints applied to botanical substances and other data-poor chemicals will be beneficial for its use in hazard identification and the characterisation of chemicals. In addition, datasets from data-rich chemicals can be used to further validate the software and illustrate the relevance, reliability and specificity of the read-across predictions within a weight of evidence framework [2,23]. 

## Figures and Tables

**Figure 1 molecules-27-06605-f001:**
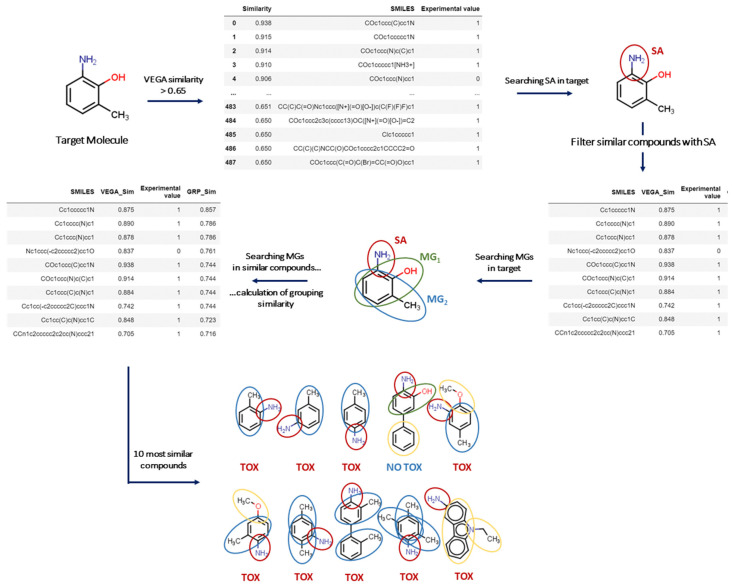
Case Study—An overall scheme of the Virtual Extensive Read-Across (VERA) algorithm. Given a target, VERA searches for similar compounds according to the Virtual models for Evaluating the properties of chemicals within a Global Architecture (VEGA) similarity index (>0.65) and the presence of structural alerts (SA, red circle) if the target substance contains any; if all similar substances are concordant in experimental values (toxic or NON-toxic), the target substance will be labelled accordingly. In the figure, molecular groups (MGs) in common between the target and similar compounds (MG_1_, green circle; MG_2_, blue circle) and not in common (yellow circles) are highlighted.

**Figure 2 molecules-27-06605-f002:**
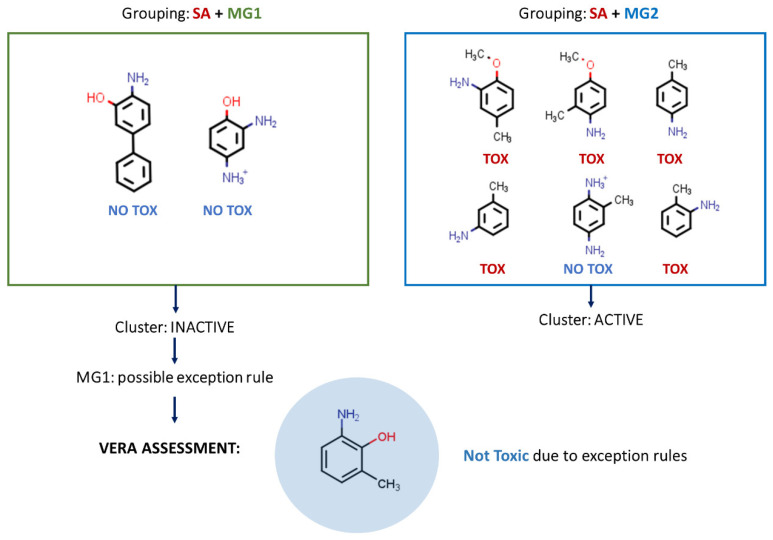
Reasoning within the VERA algorithm. For the SA in the target molecule, the algorithm analyses the co-presence of SA and MGs for similar compounds to assess the toxicity of the target and searches for any exception rules that may modulate toxicity. In the example of the figure, VERA found MG_1_ as an exception rule.

**Figure 3 molecules-27-06605-f003:**
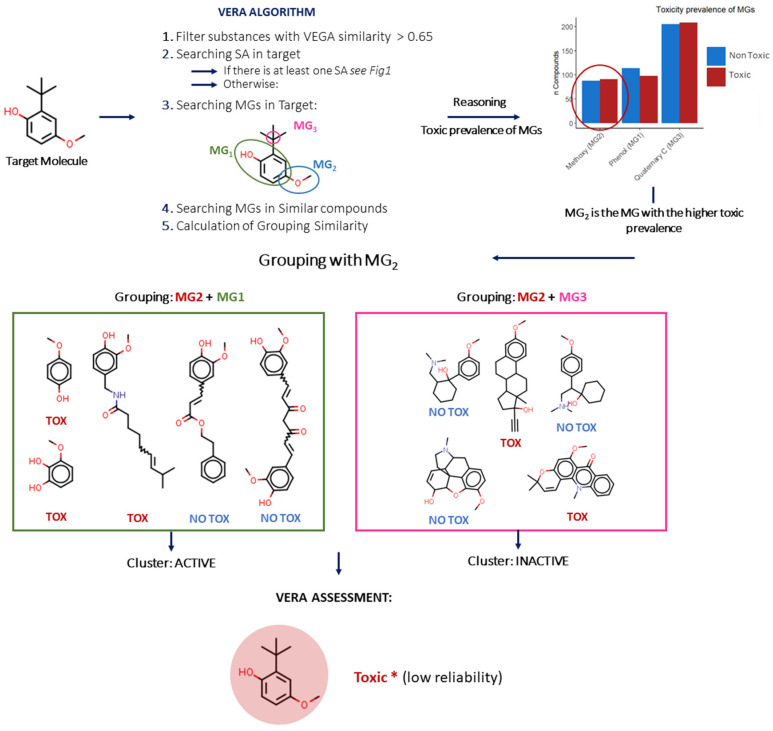
Case Study—Reasoning of the VERA algorithm. Without SA in the target molecule, VERA follows the reasoning: the algorithm analyses the co-presence of the MG with the higher toxic prevalence in the dataset and other MGs. In this case, VERA found MG_2_ as the MG with higher toxicity prevalence.

**Figure 4 molecules-27-06605-f004:**
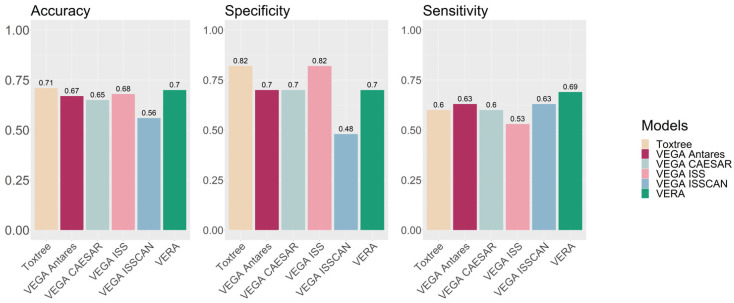
Statistical values of several QSAR models (Toxtree and VEGA for carcinogenicity) compared with VERA results.

**Table 1 molecules-27-06605-t001:** Classification parameters and the confusion matrix of botanicals.

Sensitivity		0.69	
Specificity		0.70	
Accuracy		0.69	
Precision		0.69	
**Real/predicted**	**Carcinogen**	**Non-Carcinogen**	**Equivocal**
Carcinogen	18	8	4
Non-Carcinogen	8	19	6

## Data Availability

Carcinogenicity datasets are available from www.vegahub.eu, accessed on 21 August 2022, associated to each in silico models for carcinogenicity.

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
