# Peer review of "Virtual Extensive Read-Across: A New Open-Access Software for Chemical Read-Across and Its Application to the Carcinogenicity Assessment of Botanicals"

_molecules, 2022, doi:10.3390/molecules27196605_

Round 1

Reviewer 1 Report

The manuscript entitled "Virtual Extensive Read-Across: a new open-access software for chemical read-across and its application to the carcinogenicity assessment of botanicals" presents an open-access software VERA (Virtual Extensive Read-Across) that provides a means to assess the similarity between chemicals using structural alerts specific to the 16 property, pre-defined molecular groups, and structural similarity. This software is suitable for similar substances. Additionally, the authors tested the software using carcinogenicity as a case study.  According to the present results, the software correctly labeled 70% of the botanicals, indicating the most similar substances and the main features associated with carcinogenicity.

The manuscript can be accepted. There are a few points to clarify.

1 - Abstract section

The main point in the manuscript  and abstract section is: Virtual Extensive Read-Across: a new open-access software

But I didn't realize the link access to the VERA software.

Please, add the link where one can access the software in the abstract and some key points in the manuscript (including the conclusion section),. If there isn't direct access to VERA, it seems an algorithm and not software, in this last case I propose changing the title.

2 -  Figure 1 - captions 

"and the presence of SA"

SA is an acronym previously mentioned in the text but is suitable to add its meaning again here.

3 - Figure 1 - captions 

Please add the meaning of the yellow circles.

4 - Table 1 - 

Please check the value of precision. I think that it is 0.69.

Reviewer 2 Report

The paper “Virtual Extensive Read-Across: a new open-access software for chemical read-across and its application to the carcinogenicity assessment of botanicals” is devoted to an interesting and actual problem and is of great methodological importance.

There are no principled restrictions for the publication of the manuscript, but there are some remarks:

1. In different parts of the manuscript, the authors compare the “read-across” and “in silico” methods, this is methodologically incorrect. “in silico” includes all approaches that have a computer implementation, including “read-across”. Obviously, the authors wanted to compare the “read-across” and “QSAR” methods - this corresponds to the meaning of this work.

2. In addition to a number of different useful features, the VERA tool is also intended for predicting the activity classes of the studied substances. To assess the effectiveness of VERA in this regard, a comparative analysis of its application with other classification approaches should be carried out, given that the results presented in Table 1 are not impressive.

3. In equation 1 (line 307), it is probably necessary to comment on the scheme for calculating the GSI parameter, in particular, the presence of numbers 2 and 8 in the denominators of the expression.

Thus, the manuscript can be published in "Molecules" after taking into account these comments.
